# The Beneficial Effects of GLP-1 Receptor Agonists Other than Their Anti-Diabetic and Anti-Obesity Properties

**DOI:** 10.3390/medicina61010017

**Published:** 2024-12-26

**Authors:** Chenqi Lu, Cong Xu, Jun Yang

**Affiliations:** 1Institute of Organ Transplantation, Tongji Hospital, Tongji Medical College, Huazhong University of Science and Technology, Key Laboratory of Organ Transplantation, Ministry of Education, NHC Key Laboratory of Organ Transplantation, Key Laboratory of Organ Transplantation, Chinese Academy of Medical Sciences, Wuhan 430030, China; lcq974304766@163.com; 2Division of Nephrology, Department of Internal Medicine, Tongji Hospital, Tongji Medical College, Huazhong University of Science and Technology, Wuhan 430030, China; 18535408407@163.com

**Keywords:** GLP-1, GLP-1 receptor agonists, neuroprotection, cardiovascular protection, liver protection, kidney protection, neoplasms

## Abstract

As an incretin hormone, Glucagon-like peptide-1 (GLP-1) has obvious effects on blood glucose regulation and weight loss. GLP-1 receptor (GLP-1R) agonists are synthetic products that have similar effects to GLP-1 but are less prone to degradation, and they are widely used in the treatment of type 2 diabetes and obesity. In recent years, different beneficial effects of GLP-1R agonists were discovered, such as reducing ischemia-reperfusion injury, improving the function of various organs, alleviating substance use disorder, affecting tumorigenesis, regulating bone metabolism, changing gut microbiota composition, and prolonging graft survival. Therefore, GLP-1R agonists have great potential for clinical application in various diseases. Here, we briefly summarized the beneficial effects of GLP-1R agonists other than the anti-diabetic and anti-obesity effects.

## 1. Introduction

GLP-1 is an incretin hormone, which shares approximately 50% homology with glucagon. It is mainly released from the L cells in the intestine. GLP-1 can enhance the secretion of insulin and suppress the release of glucagon in a glucose-dependent manner by activating its specific receptor (GLP-1R). Additionally, it also can reduce the apoptosis of β cells and promote their proliferation [1,2]. Therefore, GLP-1 can be a highly effective hypoglycemic agent to maintain glucose stability. GLP-1R agonists are a class of drugs designed based on the structure of exenatide or GLP-1, which can activate GLP-1R and exert effects similar to those of endogenous GLP-1. At present, GLP-1R agonists are mainly used in clinical practice for the treatment of type 2 diabetes mellitus (T2DM) and obesity and have become a new class of anti-diabetic drugs [3]. However, recent studies have expanded the understanding of GLP-1R agonists beyond their anti-diabetic and anti-obesity effects. Emerging evidence suggests that these agents may also confer beneficial effects on various other diseases, including cardiovascular, renal, liver, and cerebral diseases, dependent or independent of their hypoglycemic effects [4,5,6,7,8,9,10,11].

In this review, we aim to summarize the potential beneficial effects of GLP-1R agonists beyond their established roles in diabetes and obesity management. By exploring the latest research findings, we hope to highlight the broader therapeutic implications of GLP-1R agonists in various disease contexts, paving the way for future investigations and clinical applications.

## 2. GLP-1R Agonists

Although GLP-1 has a highly effective blood glucose-lowering effect, its half-life is very short, lasting less than 2 min, which limits its application as a hypoglycemic agent in clinical practice. The native GLP-1 produced by the human body is rapidly degraded to GLP-1 (9-37) or GLP-1 (9-36)NH2 by dipeptidyl1 peptidase-(DPP-4), which is a ubiquitous proteolytic enzyme in the cell-surface membrane or in the circulation, and then GLP-1 is inactivated [12,13,14,15]. To address this problem, GLP-1R agonists have been developed, such as exenatide, liraglutide, albiglutide, dulaglutide, semaglutide, and so on. They are less sensitive to DPP-4 than GLP-1 and thus have a longer half-life [16,17,18]. Therefore, they are more clinically useful than GLP-1.

GLP-1R agonists are the products of GLP-1 structure modification, which have both the pharmacological functions of GLP-1 and longer drug half-lives. Based on the duration of action and frequency of administration, GLP-1R agonists can be classified into short-acting (such as exenatide and lixisenatide) and long-acting (such as liraglutide, dulaglutide, albiglutide, exenatide long-acting release, and semaglutide) agonists [4]. Short-acting GLP-1R agonists markedly slow the gastric emptying, resulting in decreased glucose absorption and thus primarily lowering postprandial glucose levels, whereas long-acting GLP-1R agonists continuously enhance insulin secretion and decrease glucagon levels, and thus significantly reducing fasting plasma glucose [19,20]. Previously, GLP-1R agonists were thought to exert their effects by activating GLP-1R; however, recent studies have shown that GLP-1R agonists can also act through GLP-1R-independent pathways [21,22,23]. This GLP-1R-independent pathway effect may be mediated by the presence of alternative receptors, GLP-1 metabolites such as GLP-1 (9-36), or other potentially non-classical mechanisms (Figure 1) [24,25]. Studies have found that GLP-1R agonists still exhibited some biological effects in GLP-1R knockout (KO) mice and in tissues with low or absent GLP-1R expression, such as liver, suggesting their GLP-1R-independent functions [22,23,26,27]. Although GLP-1R is absent in GLP-1R KO mice and in the liver, there may be other members of the glucagon receptor superfamily present, such as the glucagon receptor and the GIP receptor, which share structural similarities with GLP-1R [28,29]. There is also a certain structural similarity between GLP-1 and ligands such as GIP and glucagon [30]. Therefore, some researchers believe that GLP-1 and its analogs may exert their GLP-1R-independent effects by acting on these structurally similar receptors [24]. Additionally, the metabolites of GLP-1 also appear to exert their biological functions independently of GLP-1R. For example, in GLP-1R KO mice, GLP-1 (9-37) and GLP-1 (28-37) still exhibited cardioprotective and nephroprotective effects [23,31,32]. Therefore, a portion of the GLP-1R-independent effects of GLP-1 and its analogs may be mediated by their metabolic products. There are also some speculations regarding the specific mechanisms by which these metabolites exert their effects. For example, GLP-1 (9-37) may exert its effects by binding to CD36, a transport receptor, and being transported into cells [33]. Additionally, because of its amphiphilic structure, GLP-1 (9-37) may be able to directly enter cells and exert its effects without the other receptors [34]. Some studies have also found that GLP-1 may act on sensory neurons expressing GLP-1R and subsequently exert biological effects on other organs through neural conduction [25]. This phenomenon might explain why GLP-1 can affect organs with low or absent GLP-1R expression. However, these speculations need further validation, and the exact mechanisms still need further investigation.

### 2.1. Exenatide

Exendin-4 is a GLP-1 analog isolated from the venom of Heloderma. It contains 39 amino acid peptides and shares approximately 53% sequence similarity with the native GLP-1 of human [36]. Thus, it can exert its effects by activating the GLP-1 receptor. Because it contains a Gly at position 2, it is not easily degraded by DPP-4 and thus has a longer half-life than GLP-1, which is about 2.4 h [37,38]. Exenatide is a synthetic product of exendin-4. According to some clinical studies, Exenatide can significantly reduce the level of HbAlc, improve the function of β-cell, and rapidly reduce both fasting and postprandial glucose concentration via injections twice a day [39,40]. Exenatide, as a short-acting GLP-1R agonist, was approved to treat T2DM in the US in 2005.

### 2.2. Liraglutide

Liraglutide, as a long-acting GLP-1R agonist, is widely used in the treatment of T2DM. It shares 97% sequence homology with the native GLP-1 in humans and can bind to GLP-1R to exert its effects. After a fatty acid chain was attached to the lysine at position 26 and the lysine at position 34 was substituted with arginine in the GLP-1 peptide chain, liraglutide was formed [41]. GLP-1 can bind to plasma albumin after the modification by a fatty acid chain, and the albumin will protect GLP-1 from rapid DPP-4 degradation. Thus, liraglutide has a long half-life of about 13 h and only needs once injection a day [42].

In addition to lowering blood glucose, liraglutide also can suppress appetite and reduce energy intake, which is beneficial for patients to loss weight. Therefore, liraglutide is approved for the treatment of obesity [41]. What is more, liraglutide has also shown other beneficial effects in some animal studies. For example, liraglutide plays an important role in cardioprotection and neuroprotection, improving the outcome of cardiovascular and nervous system diseases [5,43]. Additionally, its protective effect on CKD has also been reported [4]. Liraglutide was approved by FDA in 2010.

### 2.3. Semaglutide

As daily injections are too frequent for patients, which is not conductive to treatment adherence, it is necessary to develop a GLP-1 analog with longer dosing intervals. Semaglutide is a long-acting GLP-1R agonist which can be administered once a week [44]. Structurally, semaglutide is formed by the substitution of Aib for Ala at position 8, Arg for Lys at position 34, and acylation of Lys at position 26 in the GLP-1 peptide chain [44]. After the modification, semaglutide has a longer half-life, higher GLP-1R affinity, and less renal excretion than liraglutide [44]. The efficacy of semaglutide has been demonstrated by some animal and clinical studies. In addition to hypoglycemic effects, semaglutide also has shown its efficacy in cardioprotection and weight loss [45,46]. Semaglutide was approved to treat T2DM in the US in 2017.

### 2.4. Other GLP-1R Agonists

Other GLP-1R agonists include lixisennatide, dulaglutide, albiglutide, and so on. Lixisennatide belongs to short-acting GLP-1R agonists. Structurally, lixisennatide is formed on the basis of exenatide structure by removing the proline at position 38 and connecting six lysines to the serine at position 39 [47]. Dulaglutide and albiglutide are long-acting GLP-1R agonists. Albiglutide is formed by the substitution of glycine for alanine at position 8 in the GLP-1 peptide chain and the fusion of two modified chains to human plasma albumin [48]. And dulaglutide is formed by the substitution of glycine for alanine at position 8, glutamate for glycine at position 22, glycine for arginine at position 36 in the GLP-1 peptide chain, and then the fusion of the peptide chain to the constant region of human immunoglobulin G4 [49]. All of them have longer half-lives than the native GLP-1 in humans. The comparison of several GLP-1 receptor agonists is shown in Table 1.

## 3. The Role of GLP-1R Agonists in Various Diseases

### 3.1. Ischemia and Reperfusion Injury (IRI)

Ischemia and reperfusion injury refer to the damage caused by the insufficient blood supply to tissues and the subsequent restoration of blood perfusion. During the period of ischemia, due to insufficient oxygen supply, the intracellular ATP production is reduced and the lactate production is increased, which leads to intracellular acidosis, lysosomal enzyme leakage, breakdown of the cytoskeleton, and inhibition of the Na+/K+ATPase activity; then, the cell develops an intracellular accumulation of Na+ ion, Ca2+, and water, which will lead to cellular edema and the production of small amounts of ROS [75]. During the period of reperfusion, the level of oxygen increases, and the PH normalizes, which leads to the production of large amounts of ROS and the destruction of antioxidant capacity. Additionally, the level of Ca2+ further increases, which leads to the activation of calpains [75]. These result in injury of the cell cytoskeleton, the membranes, and DNA, which contributes to cell death. Following the cell death, some substances called DAMPs are released into the extracellular space, resulting in the activation of the innate and adaptive immune system [75]. Ischemia and reperfusion injury are closely related to many diseases, such as myocardial infarction (MI) and ischemic stroke, and are very common in organ transplantation. In the past decade, GLP-1R agonists have been reported to have a certain protective effect on IRI.

#### 3.1.1. Myocardial Ischemia and Reperfusion Injury (MIRI)

Pretreatment with liraglutide at a dose of 200 μg/kg improved the survival and reduced infarct size and cardiac rupture in mice after myocardial infarction and alleviated the IRI in isolated hearts [76]. Administering exendin-4 at a dose of 100 μg/kg following MI in mice reduced scar size in the heart and inhibited cardiac hypotrophy and cardiac interstitial fibrosis, thereby alleviating the cardiac remodeling and promoting angiogenesis in the infarcted myocardium [77]. In studies on rats, lixisenatide and albiglutide could alleviate myocardial ischemia and reperfusion injury and reduce infarct size [78,79]. Additionally, GLP-1 was found to be associated with the cardioprotection of remote ischemic preconditioning (RIPre) in the rat. RIPre reduced about 50% infarct size, whereas the administering of GLP-1R antagonist, exendin (9-39), abolished this effect [80]. In an in vitro study, the pretreatment with 100 nM liraglutide significantly alleviated the damage induced by hypoxia-reoxygenation in HL-1 cells, which is a cardiac muscle cell line [81]. In a clinical study, 96 patients with ST-segment elevation who had undergone urgent primary percutaneous coronary intervention (PCI) were randomized to two groups. And the group receiving subcutaneous liraglutide had better cardiac function and less infarct size at the 3 months after surgery than the group receiving placebo, which suggested that liraglutide has great potential in the prevention and treatment of MIRI [82].

GLP-1R, as a G protein-coupled receptor, has been found to activate various pro-survival pathways in animal studies [76]. When GLP-1R is activated, the function of adenylyl cyclase increases, leading to the increase in intracellular cAMP content, which elevates the phosphorylation level of MKK3, AKT, GSK3β, and other signaling molecules; after that, the expression of many protective factors increases, such as Nrf2 and HO-1, which may exert an effect of cytoprotection [76,77,83]. GLP-1R agonists, such as exenatide, can limit the production of ROS, improve mitochondrial function, and reduce the activation of caspase-3 and caspase-9, which contributes to reduced apoptosis [84]. Additionally, GLP-1R agonists have been shown to alter glucose utilization in cardiomyocytes, enhancing the uptake of glucose and increasing the oxidation of both glucose and lactate, which may be beneficial for IRI [78,85,86].

#### 3.1.2. Cerebral Ischemia and Reperfusion Injury (CIRI)

Lixisenatide at doses of 1 and 10 nmol/kg reduced infarct size and improved neurobehavioral function in a rat model of global cerebral ischemia and reperfusion injury [87]. After the administration of lixisenatide, the indicators of oxidative stress, markers of apoptosis, and inflammatory factors were all reduced. However, these protective effects were only partially reduced with the GLP-1R antagonist, exendin (9-39), suggesting that lixisenatide may exert its protection on CIRI through both GLP-1R-dependent and independent pathways [87]. In a study on acute cerebral ischemia in rats, both liraglutide and semaglutide demonstrated protective effects, reducing infarct size and improving neuroscore in a dose-dependent manner [88]. In a rat model of transient middle cerebral artery occlusion, intracerebroventricular injection of exendin-4 significantly reduced the infarct size and alleviated the inflammatory response mediated by the JNK pathway [89]. Additionally, in a gerbil model of transient cerebral ischemia, exenatide also reduced neuronal death and the activation of microglial in a dose-dependent manner by up-regulating GLP-1R expression in astrocytes and GABAergic interneurons [90]. AND in a model of CIRI in mice, the positive allosteric stimulation of GLP-1R by P7C3 activated the β-catenin, which is associated with Akt/GSK3 and cAMP/PKA, leading to the increased expression of neurogenesis proteins [91]. What is more, besides GLP-1R agonists, GLP-1 (9-36), the cleavage product of GLP-1, could also alleviate neuroinflammation in a stroke model of mice by activating the insulin-like growth factor 1 receptor [92]. The above studies indicate that GLP-1R agonists can effectively reduce CIRI and play a protective role in neurological function.

#### 3.1.3. Hepatic Ischemia and Reperfusion Injury (HIRI)

Liraglutide has shown protective effects against HIRI in both mice and rats [27,93]. In mice, the pretreatment of liraglutide at a dose of 200 μg/kg significantly reduced the increased levels of alanine aminotransferase (ALT) and aspartate aminotransferase (AST) and the levels of various inflammatory cytokines, such as TNF-α, IL-1β, and IL-6 [27]. The administering of liraglutide inhibited the polarization of macrophages to M1, an inflammatory phenotype in macrophages, which may be mediated by GLP-1R [27]. However, the protective of liraglutide on HIRI was only partially diminished in GLP1R^−/−^ mice, which suggests that there may be a GLP-1R-independent protective effect [27]. In rats, a 50 μg/kg dose of liraglutide inhibited inflammation and apoptosis and enhanced the antioxidant capacity, thereby alleviating HIRI [93]. Exendin-4 showed a protective effect on HIRI in high-fat diet (HFD)-fed mice [94]. Exendin-4 led to decreased expression of autophagy-associated proteins, such as LCII, beclin-1, p62, and high-mobility group protein B1 (HMGB1), which resulted in the suppression of autophagy and reduced the injury. This protective effect was reversed by exendin (9-39) [94].

#### 3.1.4. Other Ischemia and Reperfusion Injury

GLP-1R agonists also have a protective effect on ischemia and reperfusion injury in other organs. In a lethal renal IRI model of mice, treatment with liraglutide strongly improved the survival of mice (100% vs. 0%) with better renal function, milder pathological damage, and lower levels of inflammatory cytokines [22]. Liraglutide treatment significantly reduced the translocation of HMGB1 from the nucleus to the cytoplasm and inhibited the release of it to the extracellular space. The effect of liraglutide was partially reversed with exendin (9-39) or in GLP-1R^−/−^ mice [22]. In a rat renal IRI model, pretreatment with exenatide alleviated tissue damage and cell apoptosis by up-regulating the expression of HO-1 [95]. Moreover, in studies on rodents, GLP-1R agonists could reduce intestinal and gastric ischemia and reperfusion injury [96,97]. Pretreatment with liraglutide exerted a protective effect on intestinal IRI in mice by reducing inflammation and apoptosis via the PI3K/AKT and NF-κB pathway [97]. And liraglutide pretreatment protects against gastric IRI in rats by decreasing apoptosis and oxidative stress [96].

### 3.2. Nonalcoholic Fatty Liver Disease (NAFLD)

NAFLD, the most common reason for chronic liver disease, is a metabolic stress-induced liver injury closely associated with insulin resistance, obesity, and hypertension, which is characterized by fat accumulation and hepatic parenchymal cellular fatty lesions [98,99]. NAFLD includes nonalcoholic fatty liver (NAFL) and nonalcoholic steatohepatitis (NASH) and has a tendency to cirrhosis and liver cancer. Oversupply or impaired metabolism of fatty acids caused by various factors may lead to the production and accumulation of lipotoxic substances, which can lead to endoplasmic reticulum stress and hepatocyte injury [100]. NAFL is defined with hepatic steatosis alone, and NASH is defined with inflammation and more serious hepatocyte damage [100]. The global prevalence of NAFLD is as high as 25%, and when it develops into NASH, cirrhosis, and even liver cancer, it will seriously affect the health and quality of life of patients [101]. However, there are no efficient treatments for NAFLD currently. In recent studies, GLP-1R agonists were revealed with a beneficial effect on NAFLD through their ability to regulate metabolism, indicating that they may be a new choice for the treatment of NAFLD.

In animal models of NAFLD, GLP-1R agonists have been reported to reduce hepatic fat accumulation and alleviate hepatic damage by improving insulin resistance, adjusting lipid metabolism, and alleviating liver inflammation and oxidative stress. Liraglutide treatment significantly reduced the serum lipid profile with decreased expression of the lipogenic gene via the SHP1/AMPK pathway in mice, and it exerted antioxidant and anti-inflammation effects on NAFLD in rats and diabetic mice [102,103,104]. Additionally, liraglutide also inhibited the formation of inflammasome and the activation of pyroptosis by enhancing mitophagy, thereby significantly reducing the inflammation in NAFLD. And the same result was seen with exenatide in another study [105,106]. And in a recent study, liraglutide could activate autophagic flux via the retinoic acid receptor-related orphan receptor, leading to decreased lipid deposition in hepatocytes [107]. Exenatide could attenuate NAFLD progression by improving insulin resistance and suppressing inflammation in male rats and could reverse the lipid accumulation and the increased level of inflammation in a rabbit NAFLD model, which was abolished by PI3K inhibitors [108,109]. What is more, semaglutide also had a protective effect on NAFLD. Niu et al. found that semaglutide treatment significantly reduced body weight, LDL, TG, and inflammation cytokines in NAFLD mice. Meanwhile, the antioxidant capacity and mitochondrial morphology were improved [110]. These indicated that semaglutide has an inhibitory effect on liver fat accumulation and inflammation [110]. Cotadutide is a GLP-1R/GcgR agonist. In a study on mice, Cotadutide showed an inhibitory role in lipogenic, fibrosis, and inflammation in NASH [111].

GLP-1R agonists have also shown protective effects against NAFLD in clinical studies. Guo et al. assigned 96 patients with T2DM and NAFLD into three groups, receiving liraglutide, insulin glargine, and placebo, respectively. After 26 weeks, the treatment with both liraglutide and insulin glargine significantly reduced the content of lipids in the liver, abdomen, and viscera and improved liver function compared to placebo. Moreover, the protective effect of liraglutide is better than insulin glargine [112]. Another study reached a similar conclusion that liraglutide treatment decreased the levels of ALT and AST and improved the glucose and lipid metabolism in patients with T2DM and NAFLD [113]. In a 24-week clinical trial, 76 patients with T2DM and NAFLD were randomly assigned into two groups, receiving exenatide or insulin glargine, and the result was that both exenatide and insulin glargine had beneficial effects on the treatment of NAFLD via alleviating disorders of glucose and lipid, and the effect of exenatide is larger than insulin glargine [114]. And in another 104-week clinical study involving 695 patients with T2DM and NAFLD, both the treatment with exenatide or dapagliflozin alone and the treatment with their combination improved the hepatic steatosis and fibrosis, with the combination having the best effect [115]. In some randomized trials and clinical studies, semaglutide could reduce liver steatosis and improve the quality of life in patients with NAFLD, but it seemed to have no effect on the progression of fibrosis [116,117,118,119]. What is more, dulaglutide also improved liver function and lipid metabolism in patients with NAFLD [120].

### 3.3. Neurodegenerative Diseases

Neurodegenerative diseases, mainly including Alzheimer’s disease, Huntington’s disease, Parkinson’s disease, and so on, are irreversible neurological disorders caused by the progressive loss or dysfunction of the selectively vulnerable neurons and are characterized by the deposition of massive misfolded proteins [121,122]. The causes of neurodegenerative diseases may include oxidative stress, programmed cell death, neuroinflammation, proteotoxic stress, and mitochondrial dysfunction [123]. Additionally, metabolic disorders and T2DM are also closely related to the occurrence of neurodegenerative diseases [124]. Studies have shown that the brain expresses GLP-1R and GLP-1R agonists, which can regulate metabolic disorders and have the ability to access the blood–brain barrier [125]. Therefore, GLP-1R agonists may be a potential therapeutic agent for the treatment of neurodegenerative diseases.

#### 3.3.1. Alzheimer’s Disease (AD)

Alzheimer’s disease is a neurodegenerative disease with two characteristics: neurofibrillary tangles of hyperphosphorylated tau and β-amyloid plaque deposition [126]. The clinical manifestations are cognitive decline and behavioral disorders. The 4-week treatment of exenatide downregulated the expression of GnT-III via the Akt/GSK-3β/β-catenin pathway, resulting in fewer neuropathological changes and memory deficits in mouse Alzheimer’s disease models [127]. Liraglutide could cross the blood–brain barrier, reduce the number of β-amyloid plaques, neuroinflammation, and synapse loss, and improve memory impairment in APP/PS1 mice models of AD [128]. Similar results were also shown in both the sporadic AD model and the 5×FAD transgenic AD model of mice [129]. In a 3xTg mice model of AD, semaglutide promoted glycolysis, ameliorated memory impairment, and reduced β-amyloid plaques and neurofibrillary tangles via GLP-1R/SIRT1/GLUT4 pathway, resulting in better metabolism balance and cognition [130]. Additionally, lixisenatide and dulaglutide also reduced the neurofibrillary tangles and β-amyloid plaque deposition and, thus, alleviated the neuroinflammation and neuronal injury [131,132].

In a clinical study, exenatide seemed to have no beneficial effect on the progression of Alzheimer’s disease, while this conclusion may not necessarily be true because of the early termination of the trial [133]. In a clinical study with people at high risk of AD, liraglutide may protect the brain against glucose dysregulation and delay the progression of AD [134]. And in another study with 38 AD patients, liraglutide significantly increased the capacity of blood–brain glucose transfer, which was related to delaying the progression of AD [135]. In a recently published clinical study, researchers found that once-weekly treatment with exenatide significantly reduced the levels of AD-related inflammatory proteins in patients with AD, such as soluble vascular cell adhesion protein 1 (sVCAM-1) and plasminogen activator inhibitor 1 (PAI-1) [136]. However, in this study, the researchers did not analyze the changes in the patient’s cognitive function. In another recent clinical study, researchers analyzed the effects of once-weekly exenatide treatment on the cognitive abilities of patients with mild cognitive impairment. However, unfortunately, the once-weekly exenatide treatment did not appear to have a beneficial effect on the patients’ cognitive performance [137]. This phenomenon may be related to the progression of the disease in patients, or it could be that exenatide can only alleviate the neuroinflammation in AD patients without improving their cognitive impairment symptoms. More clinical studies are needed to further explore the effects of GLP-1R agonists on patients with AD.

#### 3.3.2. Parkinson’s Disease (PD)

Parkinson’s disease is a neurodegenerative disease characterized by the depigmentation of the substantia and the dopaminergic neurons loss in the substantia nigra pars compacta [138]. Tremor and bradykinesia are the main manifestations of PD [138]. GLP-1R agonists have been reported to have potential protective effects on endogenous dopaminergic neurons. In a study with the MPTP-induced model of Parkinson’s disease in mice, Exenatide treatment reduced the degeneration of dopaminergic neurons and preserved dopamine levels [139]. And in another study with a rat Parkinson’s disease model, Exenatide also inhibited the loss of dopaminergic neurons and the aggregation of pathological α-synuclein by enhancing autophagy via the PI3K/Akt/mTOR pathway [140]. Tyrosine hydroxylase (TH) is a marker of dopaminergic neurons. Liraglutide could reduce neuroinflammation, increase the expression of TH, and inhibit the apoptosis of TH-positive neurons in mice Parkinson’s disease models through AMPK/NF-κB pathway or by regulating mitochondria function and autophagy [141,142]. A similar effect has also been reported for other GLP-1R agonists, such as semaglutide and lixisenatide [143,144]. Recently, in a rotenone-induced Parkinson’s disease model of mice, dulaglutide increased the expression of TH and the level of dopamine through its antioxidant and anti-inflammation capacity [145]. Additionally, GLP-1R agonists may also have a protective effect on transplanted dopaminergic neurons. Dopaminergic neuron transplantation has great potential for the treatment of PD; however, the survival rate of newly transplanted neurons is not satisfactory [146]. The treatment of exenatide and liraglutide after neuron transplantation significantly increased graft survival and improved motor function in transplanted rats, which indicated that GLP-1R agonists may be beneficial for grafted patients [146].

In two randomized controlled trials, exenatide showed beneficial effects on motor disorder symptoms and cognitive impairment in PD [147,148]. In a recent phase II clinical study, the effects of lixisenatide on motor disability in patients with PD were also evaluated. The study found that after 12 months of treatment, patients treated with lixisenatide showed slower progression of motor disability compared to those receiving a placebo, but they also experienced more gastrointestinal side effects [149]. However, in a recent clinical study on NLY01 (a longer-acting, brain-penetrant exenatide), researchers found that patients with PD receiving NLY01 treatment did not show improvements in motor or non-motor symptoms compared to those receiving a placebo. Nevertheless, a subgroup analysis in this study indicated that younger patients might experience some motor benefits [150]. Therefore, more clinical studies are needed to explore the benefits of GLP-1R agonists for patients with PD and their potential side effects.

#### 3.3.3. Huntington’s Disease (HD)

Huntington’s disease is a fully penetrant monogenic inherited neurodegenerative disease for which there is currently no effective treatment. The mutant huntingtin can lead to neuronal dysfunction and death through a variety of mechanisms, resulting in motor, cognitive, and mental behavior abnormalities [151]. GLP-1R agonists may have the ability to ameliorate the progression of Huntington’s disease due to their neuroprotective effects. In a 3-nitropropionic acid-induced HD model of rats, liraglutide alleviated the neurobehavioral abnormalities and histopathological changes with less apoptosis and higher antioxidant capacity and expression of neuroprotective molecules [152]. And in an in vitro HD model, by restoring insulin sensitivity, liraglutide rescued the apoptosis of neurons, which was caused by the overexpression of mutant huntingtin [153]. In a study about HD mice, Exenatide improved abnormal glucose metabolism and motor function, and the survival time of mice was extended [154]. However, more studies are needed to confirm the role of GLP-1R agonists in the treatment of Huntington’s disease.

#### 3.3.4. Other Neurodegenerative Diseases

GLP-R agonists may have protective effects on demyelinating diseases and amyotrophic lateral sclerosis (ALS). Liraglutide could alleviate inflammation and demyelination in experimental autoimmune encephalitis (EAE) mice by enhancing autophagy and inhibiting pyroptosis [155]. And exenatide also improved clinical symptoms in EAE mice by reducing inflammation [156]. In a recent study, NLY01, a novel GLP-1R agonist, might reduce the severity and prevent the relapsing of EAE in mice by inhibiting peripheral and CNS inflammation [157]. Additionally, exenatide improved behavioral abnormalities and reduced neuronal death, whereas liraglutide did not appear to have this effect [158,159].

### 3.4. Substance Use Disorders

Substance use disorders are abnormal patterns of substance consumption that can lead to impairment in mental functioning and behavioral patterns [160]. Recent studies have found that GLP-1R agonists may have the ability to reduce the use of addictive substances such as alcohol, tobacco, and drugs.

#### 3.4.1. Alcohol Use Disorders (AUD)

In animal models of AUD, GLP-1 and its agonists, such as Exendin-4, liraglutide, dulaglutide, and semaglutide, have been shown to significantly reduce alcohol consumption in rats [161,162]. Similar results have also been observed in studies involving primates [163]. In self-administration models, research has shown that GLP-1R agonists can inhibit mice’s motivation to seek alcohol [164]. Additionally, GLP-1R agonists may also prevent relapse drinking and alleviate withdrawal symptoms [165,166]. A clinical study found that after semaglutide treatment, all six patients with AUD showed significant improvement in AUD symptoms following weight loss, which is consistent with preclinical studies [167]. Therefore, it is believed that GLP-1R agonists may be highly promising drugs for AUD in clinical practice.

#### 3.4.2. Nicotine Use Disorders

Additionally, in nicotine use disorder models, GLP-1R agonists can improve nicotine-related behaviors. Tuesta et al. discovered that activating GLP-1 neurons chemically in the nucleus tractus solitarius could result in decreased nicotine consumption in mice [168]. Similarly, GLP-1R agonists such as exendin-4 have shown the same effect [168]. Additionally, liraglutide could reduce nicotine self-administration and the motivation to seek nicotine in rats, as well as improve nicotine-related withdrawal symptoms [169]. Arillotta et al. collected online discussions from December 2019 to June 2023 and found that the use of GLP-1R agonists might reduce nicotine intake among individuals [170]. However, in a clinical study on the effects of dulaglutide on smoking cessation, dulaglutide treatment reduced weight gain after quitting smoking but did not change the smoking cessation rate [171]. Currently, there are only few clinical studies on this topic; thus, more clinical research is needed to determine whether GLP-1R agonists are effective for smoking cessation and whether there are differences in the effects of different GLP-1R agonists.

#### 3.4.3. Opioid Use Disorders

What is more, GLP-1R agonists can also reduce drug intake in rodents, such as opioids. In rats, both exendin-4 and liraglutide treatment reduced fentanyl self-administration and the motivation to seek fentanyl while simultaneously activating neuropeptide Y2 receptors could eliminate the adverse effects of GLP-1R agonists without compromising their efficacy [172,173,174]. Additionally, treatment with exendin-4 and liraglutide also reduced heroin dependence in rats, decreasing their self-administration and motivation to seek heroin [175,176]. Activation of GLP-1R by exendin-4 inhibited self-administration and seeking behavior for oxycodone in rats, but it did not reduce the analgesic effects of oxycodone [177]. However, one study indicated that exendin-4 did not appear to affect addiction-related behaviors associated with morphine and remifentanil in mice [178]. And there is almost no research on the effects of GLP-1R agonists on cannabis addiction, with only one retrospective cohort study suggesting that semaglutide may be associated with a reduction in cannabis use [179]. In summary, preclinical research results suggest that GLP-1R agonists may be a highly promising class of drugs for treating drug addiction, but there is currently a lack of sufficient clinical studies to validate their effects.

The mechanism by which GLP-1R agonists reduce substance addiction may be related to the reward pathways in the central nervous system. The nucleus accumbens (NAc) is an important region that mediates reward effects, and an increase in dopamine (DA) release within it can produce reward effects [180]. Research has found that GLP-1R is expressed in the NAc region, and GLP-1R agonists can reduce the increase in DA induced by alcohol, nicotine, and drugs, thereby diminishing the reward effects caused by these substances [181,182,183]. Additionally, in studies on the inhibitory effects of GLP-1R agonists on nicotine intake, it was found that GLP-1R agonists could also activate the GLP-1R in the interpeduncular nucleus (IPN), leading to an increase in excitatory postsynaptic currents (EPSCs) in IPN neurons. This increase produced an aversive effect on nicotine, counteracting the reward effects associated with the substance [168]. The weakening or disappearance of the reward effects would lead to a reduction in substance intake and a decrease in the motivation to seek the substance [168,180]. However, the inhibitory effects of different GLP-1R agonists on substance use disorders also vary, which may indicate that their mechanisms of action are not entirely the same and require further investigation.

### 3.5. Neoplasms

A tumor refers to the new growth formed by the abnormal proliferation of local tissue cells under the action of various tumorigenic factors. It can be divided into benign tumors and malignant tumors, and malignant tumors seriously affect the survival and quality of life of patients. The association of GLP-1R agonists with tumors has been studied in some of the literature, but the conclusions seem to be controversial.

Some studies indicate that GLP-1R agonists may prevent the progression of tumors [184,185,186,187,188]. Liraglutide significantly suppressed hepatocarcinogenesis in mice with diabetes and NAFLD [184]. And it had a synergistic effect with docetaxel on prostate cancer cells, decreasing the viability and inducing the apoptosis of cells [185]. Exenatide inhibited the growth of hepatoma cells in vitro and in vivo through the cAMP-PKA-EGFR-STAT3 pathway [186]. Additionally, exenatide also dose-dependently attenuated the proliferation of breast cancer cells in vitro and in vivo by inhibiting the activation of NF-κB and suppressing the proliferation, invasion, migration, and epithelial-to-mesenchymal transition of glioma cells by GLP-1R/sirt3 pathway [187,188]. However, there are also some studies suggesting that GLP-1R agonists promote the formation of tumors [189,190]. For example, in a nested case–control analysis of patients with type 2 diabetes receiving drug treatment, the treatment of GLP-1R agonists increased the risk of medullary thyroid cancer and thyroid cancer [189]. And in another animal study, liraglutide treatment promoted the progression of breast cancer by activating the GLP-1R/NOX4/ROS/VEGF pathway, which was significantly reversed by exendin (9-39) [190]. Therefore, further investigation into the relationship between GLP-1R agonists and tumors is essential.

### 3.6. Atherosclerosis

Atherosclerosis is a major cause of cardiovascular disease, and its occurrence is closely related to obesity and diabetes. It is characterized by endothelial dysfunction, recruitment and infiltration of inflammatory cells into the subendothelial space, proliferation of smooth muscle cells and secretion of extracellular matrix, and the formation of foam cells by macrophages and smooth muscle cells, which phagocytose lipids [5]. Studies have found that GLP-1R agonists can decrease the risk of cardiovascular disease and alleviate the progression of atherosclerosis.

GLP-1R agonists could alleviate the inflammation of the vascular wall in atherosclerosis. In a mice atherosclerosis model, liraglutide inhibited the expression of pro-inflammatory cytokines, reduced the number of pro-inflammatory cells, and increased the number of anti-inflammatory cells, which led to the attenuation of atherosclerosis [191]. And in human aortic endothelial cells, dulaglutide suppressed the increased levels of oxidative stress and pro-inflammatory cytokines induced by oxidized low-density lipoprotein [192]. Exenatide also inhibited the recruitment and inflammatory response of macrophages in atherosclerosis models [193]. Additionally, GLP-1R could reduce the formation of foam cells. Liraglutide had a significant inhibitory effect on the formation of macrophage foam cells both in vitro and in vivo, and the same effect was also observed in the treatment of exenatide and native GLP-1 [194,195]. What is more, GLP-1R agonists could inhibit the rupture of atheromas. Matrix metalloproteinases (MMP) and their inhibitors, tissue inhibitors of metalloproteinases (TIMP), are the key molecules for plaque stabilization. Semaglutide and exenatide upregulated TIMP and downregulated MMP, resulting in a lesser rupture of atheromas [196,197].

The protective effect of GLP-1R agonists on atherosclerosis has also been confirmed in clinical studies. In T2DM patients, liraglutide treatment improved lipoprotein and lipid abnormalities, suppressed vascular inflammation, and decreased adhesion molecule expression and intima-media thickness [198,199,200,201]. In a randomized controlled trial with 66 T2DM patients, the progression of atherosclerosis has been prevented by exenatide during the 52-week treatment period [202]. However, in another 18-month clinical study, the plaque volume and composition in carotid were not changed by exenatide, and the reason required further investigation [203]. The clinical study about the role of semaglutide in the progression of atherosclerosis is ongoing [204].

### 3.7. Polycystic Ovary Syndrome (PCOS)

PCOS, the most common endocrine disease in women of reproductive age, is the main cause of anovulatory infertility. PCOS is closely associated with obesity and insulin resistance (IR), and the manifestations of PCOS mainly include hyperandrogenism (HA), polycystic ovary, oligomenorrhea, and anovulation [205]. Given the association between obesity, IR, and PCOS, GLP-1R agonists were considered in the treatment of PCOS, and studies have shown that GLP-1R agonists do have therapeutic effects on PCOS.

Firstly, GLP-1R agonists reduced body weight and obesity in PCOS. In the dehydroepiandrosterone (DHEA)-induced PCOS model of rats, both exenatide and liraglutide significantly reduced the body weight and fat mass, which might be related to the reduced food intake and the improvement of IR and dyslipidemia (more triglycerides and less HDL cholesterol) [206,207]. In clinical studies, combination treatment with liraglutide, metformin, and lifestyle intervention significantly reduced the body weight in obese patients with PCOS [208]. Additionally, liraglutide and dulaglutide also had effects on the loss of visceral adipose tissue in PCOS patients [209,210]. Secondly, GLP-1R agonists improved IR and HA. In a rat PCOS model, exenatide decreased serum androgen and the homeostasis model of IR (HOMA-IR) by activating the AMPK/SIRT1 pathway [211]. And in a mice PCOS model, liraglutide and semaglutide improved the sex hormone abnormalities and reduced the level of insulin and the values of HOMA-IR by alleviating inflammation and promoting the browning of adipose tissue [212]. In overweight patients with PCOS, the treatment of metformin combined with exenatide or liraglutide had a visible effect on reducing hyperandrogenemia and improving glucose metabolism disorders [213,214]. Thirdly, GLP-1R agonists improved the reproductive function. Exenatide significantly recovered the estrus cycle and ovarian morphology with fewer follicles and more granular cell layers in PCOS rats [206,215]. And similar effects have also been reported with liraglutide and semaglutide. After the treatment of liraglutide and semaglutide, more corpus luteum and fewer cystic follicles were observed in the ovarian tissue of PCOS mice, and the disorders of the estrous cycle were improved, which indicated that GLP-1R agonists reversed pathological abnormalities in ovarian; additionally, GLP-1R agonists also improved the abnormal expression of ovarian steroidogenic enzymes in PCOS mice [212]. In obese PCOS patients, both exenatide and liraglutide increased the pregnancy rates and improved the menstrual cycle [216,217]. Finally, GLP-1R agonists alleviated the hepatic and cardiovascular abnormalities induced by PCOS. In obese PCOS patients, liraglutide and exenatide treatment were significantly associated with better coagulation function and lower inflammation and expression of adhesion molecules, which indicated a lower risk of cardiovascular diseases [218,219]. Additionally, in a randomized controlled trial with 72 PCOS patients, liraglutide treatment reduced the prevalence of NAFLD [209]. The role of GLP-1 receptor agonists in improving PCOS is briefly illustrated in Figure 2.

### 3.8. Osteoporosis

Osteoporosis is a disorder of bone metabolism characterized by bone mass loss, decreased bone mineral density, and destruction of bone microstructure, resulting in increased bone fragility and fracture risk [220]. Diabetes, hormone changes, aging, and loss of calcium and vitamin D are the main causes of osteoporosis [220,221]. Studies have found that GLP-1R agonists play a role in promoting bone formation, inhibiting bone resorption and reducing fracture risks.

In vitro studies, GLP-1R agonists inhibited osteoclastogenesis [222]. Li et al. found that RAW264.7 and bone marrow-derived macrophage (BMDM) expressed GLP-1R; after the knockdown of GLP-1R, the RANKL-induced osteoclast formation of these cells increased, whereas liraglutide treatment reduced the osteoclast formation by inhibiting NF-κB/MAPK-NFATc1 pathway [222]. Additionally, GLP-1R agonists also promoted osteoblastogenesis [223,224,225]. By activating the IGF-1/IGF-1R pathway, Exenatide treatment increased the proliferation of senescent osteoblasts with lower aging-related gene expression and higher bone metabolism gene expression [223]. Exenatide also promoted the polarization of BMDM to M2 phenotype and increased the number of Bone marrow mesenchymal stem cells [225]. Liraglutide increased the proliferation and inhibited the apoptosis of MC3T3-E1 cells which is a preosteoblast cell line [224].

GLP-1R agonists also had protective effects in animal models of osteoporosis. In rat models of diabetic osteoporosis, liraglutide treatment inhibited bone resorption and increased bone formation, and liraglutide combined with insulin improved bone mineral density and decreased the levels of inflammatory cytokines [226,227]. In ovariectomized mice, liraglutide improved bone strength, trabecular bone mass, and architecture [228]. What is more, in glucocorticoid-induced osteoporosis models of rats, liraglutide treatment decreased bone resorption indicators and increased bone formation indicators [229].

At present, there are few clinical studies about the effects of GLP-1R agonists on osteoporosis. But, in a meta-analysis, patients treated with liraglutide had a lower fracture rate than those treated with placebo, and exenatide had the best effect, indicating that GLP-1R agonists could decrease the risk of fracture [230].

### 3.9. Kidney Disease

The kidney is an important excretory organ. It is prone to acute kidney injury (AKI) with the effects of ischemia, poisoning, or other pathogenic factors, and persistent kidney injury will develop into chronic kidney disease (CKD), eventually leading to renal failure, which seriously affects the health of patients [231]. In recent studies, GLP-1R agonists have shown some renoprotective effects.

GLP-1R agonists could alleviate the AKI induced by ischemia and nephrotoxic substances. In the renal ischemia or cisplatin-treated mice, liraglutide pretreatment significantly reduced the levels of creatinine and urea nitrogen and improved the pathological damage by inhibiting the nuclear–cytoplasmic translocation and extracellular release of HMGB1, which were partially reversed by exendin (9-39) or GLP-1R^−/−^ mice [22,23]. And in rats, the treatment of exenatide significantly ameliorated the deteriorating renal function induced by contrast through the increased antioxidant capacity and endothelial function [232]. Through the activation of the PI3K/AKT pathway, semaglutide reduced the inflammation and oxidative stress induced by renal ischemia in mice [233]. These results indicate that GLP-1R agonists may be potential drugs to alleviate AKI.

GLP-1R agonists could also slow the progression of CKD. T2DM is closely related to CKD and can lead to diabetic kidney disease (DKD). Exenatide significantly improved the high glucose (HG)-induced renal tubular injury in mice and reduced the HG-induced renal tubular epithelial cell death in vitro, suggesting that GLP-1R agonists might have a protective effect on DKD [234]. In a study with rats, a high-fat-diet-induced renal inflammation, metabolic disorders and fibrosis, and impaired mitochondrial function, whereas liraglutide treatment attenuated these changes and inhibited the lipid accumulation in renal by Sirt1/AMPK/PGC1α pathway [235]. In a clinical study with advanced DKD patients, GLP-1R agonists delayed the timing of dialysis and slowed the progression to end-stage kidney disease [236]. Similar results were reported in a meta-analysis that GLP-1R agonists reduced the incidence of renal failure in patients with T2DM [237]. Additionally, 24-week treatment of Exenatide combined with insulin glargine significantly reduced the albuminuria in T2DM patients with DKD, which provided evidence for the application of GLP-1R agonists in DKD [238].

### 3.10. Gut Microbiota

Recently, the gut microbiota has attracted a great deal of attention due to its close relationship with human health. The dysregulation of gut microbiota can contribute to a variety of diseases, such as neurodegenerative diseases, cardiovascular diseases, metabolic diseases, and so on [239]. Studies have found that there is a close association between gut microbiota and GLP-1. The dysregulated gut microbiota could affect the secretion of GLP-1 and the expression of GLP-1R, leading to GLP-1 resistance and local or systemic inflammation [240,241,242,243]. However, GLP-1R activation also seemed to affect the balance of gut microbiota in recent studies.

In a study with mice, the acute administration of liraglutide significantly increased the levels of *Escherichia coli* (*E. coli*) through the activation of sympathetic nerves, and the release of NE into the gut, and a similar result was also observed in mice with colitis that liraglutide reduced the expression of tight junction gene in the cecum, resulting in bacterial translocation [244]. In another study on mice, semaglutide changed the composition of gut microbiota with more Escherichia Shigella, and the specific composition was influenced by the GLP-1R of gut intraepithelial lymphocytes [245]. Additionally, in rodents, liraglutide improved the abnormal gut microbiota composition induced by methionine-choline deficient diet and obesity, thereby reducing liver damage and preventing weight gain [246,247]. The weight loss effect of liraglutide was also related to gut microbiota regulation [248]. In a clinical study with T2DM patients, after the treatment of duodenal mucosal resurfacing combined with liraglutide, the gut microbiota diversity was changed, leading to improvements in metabolism; however, the causality between liraglutide treatment and the changes in gut microbiota could not be proven [249]. Based on the above studies, GLP-1R agonists may affect the composition of gut microbiota and thus exert protective effects on certain diseases.

### 3.11. Organ Transplantation

Organ transplantation is a treatment option for end-stage organ diseases; however, due to the presence of transplant rejection reactions, transplanted organs rarely survive for a very long time. In recent years, several studies have found that GLP-1R agonists might have the effect of prolonging graft survival. Cardiac transplant vasculopathy is an important factor affecting the long-term survival of transplanted hearts [250]. In a study on mice, it was found that the expression of GLP-1R in transplanted hearts was significantly increased, and treatment with liraglutide alleviated cardiac transplant vasculopathy and cardiac fibrosis, thereby prolonging the survival time of the transplanted heart [250]. The lymphocyte-dominated adaptive immune system is an important participant in transplant immunity. Nasr et al. found that in mice models of islet and heart transplantation, GLP-1R-positive T cells proliferated and infiltrated the grafts, and this population was primarily composed of exhausted CD8 T cells [251]. GLP-1R might serve as a negative regulator of T cell activation, similar to PD-1, whose activation might inhibit T cell function, thereby reducing rejection responses and prolonging graft survival [251]. However, there is currently a lack of research on the effects of GLP-1R agonists on adaptive immune responses and transplant rejection. Additionally, the role of GLP-1R agonists in organ transplantation also lacks adequate clinical evidence. In addition to affecting graft survival, GLP-1R agonists have also been found to potentially alleviate the side effects of immunosuppressants, which may also support their application in organ transplantation [252].

## 4. Conclusions

As the emerging anti-diabetic and anti-obesity drugs, GLP-1R agonists are widely used in clinical practice with the effects of reducing blood glucose and body weight. However, according to recent animal studies and clinical studies, GLP-1R agonists also have other beneficial effects. For example, GLP-1R agonists can alleviate the IRI of various organs, improve the metabolic disorders in NAFLD and the endothelial dysfunction in Atherosclerosis, reduce the death of neurons in neurodegenerative diseases and the HA in PCOS, reduce the use of addictive substances, promote the bone formation in osteoporosis, delay the deterioration of renal function, regulate gut microbiota, and prolong graft survival (Figure 3). The relationship between GLP-1R agonists and tumor growth is not well defined; studies have shown that for some tumors, such as liver cancer, GLP-1R agonists inhibit their growth, but for others, such as thyroid cancer, GLP-1R agonists actually accelerate their growth.

Because of these effects, people have more and more interest in the use of GLP-1R agonists. However, until now, these beneficial effects of GLP-1R agonists have been largely attributed to their capacities of anti-oxidation, anti-inflammation, metabolic regulation, and central nervous system regulation, and few studies have reported the relationship between GLP-1R agonists and the adaptive immune system. As is known to all, adaptive immune disorder is also an important cause of diseases, so it is of great interest to explore whether GLP-1R agonists can affect the adaptive immune system. Additionally, some studies have reported that the beneficial effects of GLP-1R agonists were only partially reversed by GLP-1R antagonists or GLP-1R^−/−^ mice, indicating the existence of the GLP-1R-independent pathway, but the exact mechanism is unclear. What is more, GLP-1R agonists may exert protective effects in indirect ways, such as regulating body weight and blood glucose, or directly act on target organs, but it is still unknown which way is dominant.

Therefore, we believe that efforts are still needed to further characterize the exact mechanisms of GLP-1R agonists. Even when addressing the same disease, different GLP-1R agonists can yield distinct effects. For instance, when treating patients with AD, liraglutide may be preferred over exenatide, as clinical evidence suggests that liraglutide significantly improved cognitive function, while exenatide did not. Additionally, clinical studies suggest that liraglutide may promote breast cancer growth; therefore, its use might need to be restricted in immunosuppressed patients. Thus, we believe that selecting the most suitable GLP-1 receptor agonist based on the specific disease being treated and the characteristics of the patient population is highly valuable.

## Figures and Tables

**Figure 1 medicina-61-00017-f001:**
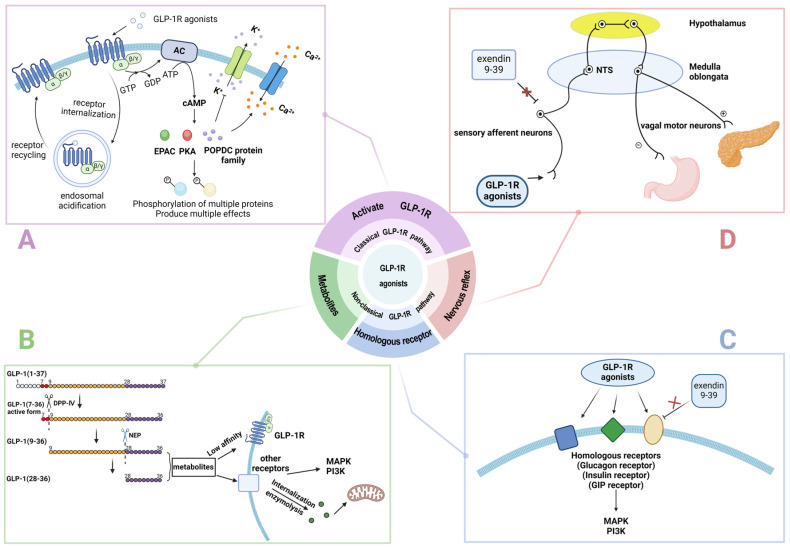
Possible mechanisms of action of GLP-1 receptor agonists. (**A**) The GLP-1 receptor (GLP-1R) is a 7-fold transmembrane G protein-coupled receptor (GPCR), which can identify GLP-1 and its analogs. GLP-1R agonists can bind to GLP-1R to form a complex, which in turn activates the G protein linked to the receptor by binding it to GTP. Then, the activated G protein activates the adenylyl cyclase (AC), which in turn catalyzes the production of cyclic adenosine monophosphate (cAMP) from adenosine triphosphate (ATP). Following an increase in cAMP levels, some proteins are activated, including protein kinase A (PKA), the Exchange Protein Activated by cAMP (EPAC), the Popeye Domain Containing (POPDC) protein family, and so on. After activation, these proteins perform a range of functions. They can exert effects on ion channels, such as promoting calcium (Ca^2+^) channel opening and potassium (K+) channel closing, and they will also catalyze the phosphorylation of corresponding downstream proteins, resulting in multiple biological effects. After exposure to GLP-1 and its analogs, GLP-1R undergoes endocytosis and accumulates in the endosomal compartments. Then, the ligand is proteolyzed by endosomal acidification, and the GLP-1R is recycled to the membrane. After these three processes, GLP-1R is desensitized and can be reactivated [35]. (**B**) GLP-1 (1-37) is cleaved to produce GLP-1 (7-36), which is the active form of GLP-1. With the action of dipeptidyl peptidase (DPP-4), GLP-1 (7-36) is cleaved to GLP-1 (9-36) and can be further cleaved to GLP-1 (28-36) in response to neutral endopeptidase (NEP). GLP-1 (9-36) and GLP-1 (28-36) are the metabolites of GLP-1, which have a very low affinity for GLP-1R. They can bind to other receptors (which are still unclear), on the one hand, to activate downstream protein kinases, such as MAPK, PI3K, etc., and on the other hand, to be internalized and cleaved to many small peptides which can affect mitochondrial function. GLP-1R is not involved in these processes [24,31]. (**C**) In addition to GLP-1R, GLP-1R agonists may also bind to some receptors which share homology with GLP-1R, such as the glucagon receptor superfamily, insulin receptor, glucose-dependent insulinotropic peptide (GIP) receptor, insulin-like growth factor 1 (IGR-1) receptor, and so on. After binding GLP-1R agonists to these receptors, downstream protein kinases such as MAPK, PI3K, etc., are activated, and this effect is not blocked by exendin 9-39, which is a GLP-1R antagonist [24]. (**D**) Peripheral GLP-1R agonists can interact with sensory afferent neurons, through which nerve impulses are transmitted to the nucleus tractus solitaries (NTS) and activate neurons in it, and then upward to the hypothalamus. After the information is integrated, the hypothalamus sends out descending impulses, which may activate the vagal motor neurons. The activated vagal motor neurons can send stimulatory or inhibitory impulses to various organs, thereby regulating organ function. For example, they can promote pancreatic secretory function and inhibit gastrointestinal motility and gastrin release. These effects, which involve afferent vagus signaling, are not blocked by exendin 9-39 [25]. This figure was created at https://BioRender.com.

**Figure 2 medicina-61-00017-f002:**
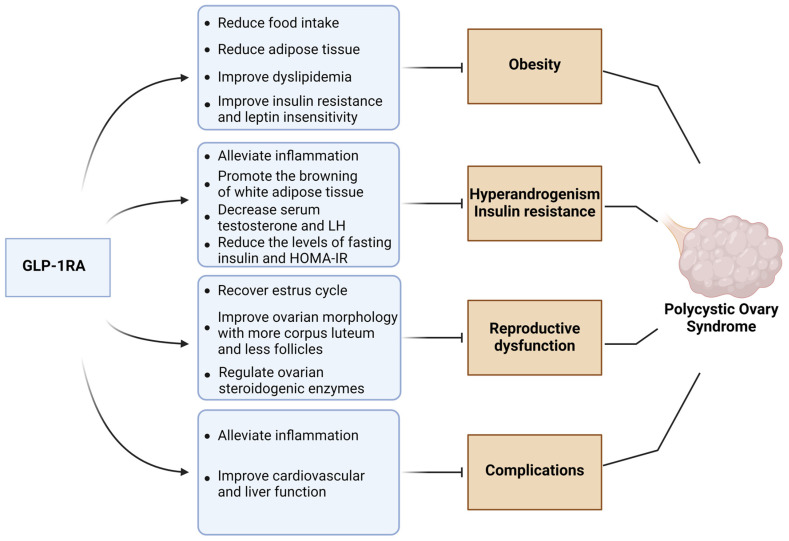
Beneficial effects of GLP-1 receptor agonists on PCOS (created in https://BioRender.com).

**Figure 3 medicina-61-00017-f003:**
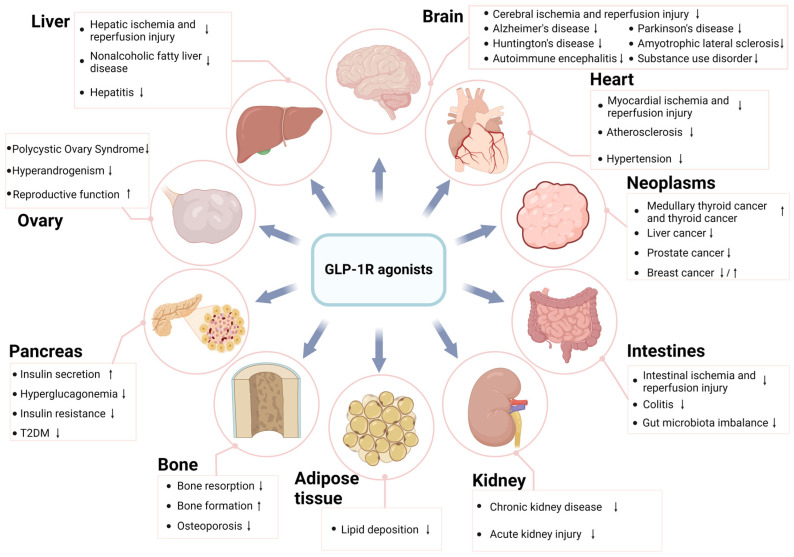
Biological effects of GLP-1 receptor agonists (created in https://BioRender.com). ↓ indicates the ability to alleviate disease damage or exert an inhibitory effect; ↑ indicates the ability to promote disease progression or exert a stimulatory effect.

**Table 1 medicina-61-00017-t001:** Comparison of several GLP-1 receptor agonists. ↓ indicates an inhibitory effect; ↓↓ indicates a stronger inhibition than ↓; √ indicates that this effect is present; × indicates that this effect is absent.

GLP-1R Agonists	Basis of Modification	Administration Route	Half-Life	Administration Frequency	Effects on Postprandial Glucose	Effects on Fasting Blood-Glucose	Effects on Gastric Emptying	Effects on Insulin Secretion	Dosage	The Side Effects of Drugs	References
Short-acting agonists											
Exenatide	exendin-4	subcutaneous injection	2.4 h	twice daily	↓↓	↓	√	reduce postprandial secretion	5 μg/dose for 1 month then up to 10 ug/dose if well tolerated	mild to moderate nausea, diarrhea, and vomiting injection site-related adverse events anti-exenatide antibody formation acute pancreatitis acute renal failure hypersensitivity reactions eosinophilia depression worsening	[50,51,52,53,54,55,56]
Lixisenatide	exendin-4	subcutaneous injection	3 h	once daily	↓↓	↓	√	reduce postprandial secretion	10 μg/dose for 14 days then up to 20 ug/dose if well tolerated	nausea and diarrhea injection site-related adverse events acute pancreatitis anaphylactic shock	[57,58,59]
Long-acting agonists											
Liraglutide	GLP-1	subcutaneous injection	13 h	once daily	↓	↓↓	×	Continuously increase	0.6 mg/dose for 1 week then up to 1.2 mg/dose further up to 1.8 mg/dose if well tolerated	nausea injection site-related adverse events acute pancreatitis gastroparesis acute kidney injury hepatotoxicity electrolyte disturbance pancreatic cancer	[60,61,62,63,64,65]
Albiglutide	GLP-1	subcutaneous injection	5 d	once weekly	↓	↓↓	×	Continuously increase	30 mg/dose	nausea, vomiting generalized edema pancreatitis injection site-related adverse events	[57,66,67]
Dulaglutide	GLP-1	subcutaneous injection	4.7 d	once weekly	↓	↓↓	×	Continuously increase	0.5–2 mg/dose	nausea, diarrhea injection site-related adverse events acute kidney injury rash	[68,69,70]
Semaglutide	GLP-1	oral/subcutaneous injection	1 w	once weekly	↓	↓↓	×	Continuously increase	0.25 mg/dose for 4 weeks then up to 0.5 mg/dose for 4 weeks then up to 1 mg/dose	mild-to-moderate nausea, dyspepsia, vomiting, headache and decreased appetite acute kidney injury injection site-related adverse events pemphigoid	[71,72,73]
Exenatide extended-release	exendin-4	subcutaneous injection	2.4 h once released	once weekly	↓	↓↓	×	Continuously increase	2 mg/dose	nausea, diarrhea, vomiting injection site-related adverse events anti-exenatide antibody formation granulomatous panniculitis	[57,74]

## Data Availability

Not applicable.

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
