# Peer review of "The Beneficial Effects of GLP-1 Receptor Agonists Other than Their Anti-Diabetic and Anti-Obesity Properties"

_medicina, 2024, doi:10.3390/medicina61010017_

Round 1

Reviewer 1 Report

Comments and Suggestions for Authors

The Beneficial Effects of GLP-1 Receptor Agonists other than 2 Their Anti-Diabetic and Anti-Obesity Properties

In this review, the authors Lu et al., have summarized the beneficial effects of GLP-1R agonists other than the anti-diabetic and anti-obesity effects. The review is extensive and well written.

Review comments:

Introduction: The narrative on GLP-1R agonists and DPP-4 may be moved to the next paragraph that discusses specifically the GLP-1R agonists. Introduction may focus more on the discovery and mechanism of action.

Substance use disorders: This section may be given subtitles as in the previous sections. E.g., Alcohol use disorders, nicotine use disorders, opioid use disorders.

In many sentences, it is not clear if the cited results are from animal studies or from human studies. Authors are requested to make it clear. It would be of great benefit to the readers to know the test subjects.

Abbreviations need to be expanded in the first use. E.g., PCI.

Grammatical and syntactical errors (italicization, spacing, etc.) are observed throughout the document that need to be corrected in proofreading.

Figure 1: The space may be better utilized. The pathway/structure illustrations within the four boxes may be redrawn bigger with bigger font for improved clarity.

Figure 2: Font size of letters within the boxes need to be increased for clarity. The circles may be aligned for symmetry.

The authors may also cite similar reviews such as

Wilbon SS, Kolonin MG. GLP1 Receptor Agonists-Effects beyond Obesity and Diabetes. Cells. 2023 Dec 28;13(1):65. doi: 10.3390/cells13010065. PMID: 38201269; PMCID: PMC10778154.

Vergès B, Bonnard C, Renard E. Beyond glucose lowering: glucagon-like peptide-1 receptor agonists, body weight and the cardiovascular system. Diabetes Metab. 2011 Dec;37(6):477-88. doi: 10.1016/j.diabet.2011.07.001. Epub 2011 Aug 25. PMID: 21871831.

Author Response

We sincerely appreciate your thorough review and constructive suggestions. We believe that your suggestions are very helpful in enhancing the quality of our manuscript. We have included our responses in the attachment.

Reviewer 2 Report

Comments and Suggestions for Authors

Objective and Key Contributions Summary:

This review intends to extensively summarize the additional beneficial properties of GLP-1 receptor agonists (GLP-1RAs) across a range of diseases beyond diabetes and obesity. These include the compilation of preclinical and clinical evidence of cardiovascular, neuroprotective, and renal benefits, with emerging roles in oncology, addiction, and perioperative organ transplantation.

General Concept Comments:

Strengths:

- The manuscript is well-written, structured, and balanced, addressing a wide range of physiological and pathological conditions.

- It cites relevant and current references to demonstrate the latest progress.

- Figures and tables (e.g., mechanisms and comparisons of GLP-1RAs) also help in understanding.

Weaknesses:

- The coverage of some sections is not very substantial, especially in a clinical implications section and a section for the translation of findings to practice.

- The section on GLP-1R-independent pathways is intriguing but lacking; providing a few examples would create some enjoyable backstory.

Completeness:

- The range of conditions discussed is broad, but the situations regarding neurodegenerative diseases could have some deeper understanding and limitations of mechanisms placed.

Potential Bias:

- Do not mention the same authors or studies over and over again. The vast majority of references are pertinent, but there are a few that could be slimmed down for brevity.

Specific Comments:

- Expand on the role of alternative receptors for GLP-1 metabolites (e.g., GPR119 and others).

- Table 1: Provide more comparative details on adverse effect profiles of short-acting vs. long-acting GLP-1RAs.

- Section 3.3 (Neurodegenerative Diseases): Recent clinical trials would be beneficial to have for increased clinical relevance in Alzheimer's and Parkinson's studies.

- Clarify “most appropriate GLP-1R agonist for each disease”; which diseases and which populations.

General Questions:

 Are the citations dated and topical?

  - Most are recent (in the last 5 years). 

- Is it suitable to use figures and tables?

  - Yes, but make sure that all legends explain the data being presented to the point that the reader does not have to refer to the main text in order to understand the data.

- Are the experiments sufficient and appropriate to address the aims?

  - Preclinical evidence is well discussed, but findings of human trials can be incorporated in a more systematic manner.

- Do the conclusions drawn correspond to the evidence you presented?

  - Mostly, but nuanced conclusions regarding tumors (Section 3.5) and addiction (Section 3.4) could do with more explicit disclaimers.

Author Response

(The authors gave the same response as above.)

Reviewer 3 Report

Comments and Suggestions for Authors

Interesting paper.  Great work.. I just have a couple of comments.

There is a typo on line 506..  Sentence says that GLP-1R agonists play a role in promoting bone formation, inhibiting bone broken and alleviating osteoporosis. 

I think you may have meant to say inhibiting bone resorption and reducing fracture risks.  I do not think there is evidence that the drugs is alleviating osteoporosis as fracture risk does not equal osteoporosis.  

Author Response

We sincerely appreciate your thorough review and constructive suggestions. We believe that your suggestions are very helpful in enhancing the quality of our manuscript.

Comment 1: There is a typo on line 506..  Sentence says that GLP-1R agonists play a role in promoting bone formation, inhibiting bone broken and alleviating osteoporosis.

Response: Thank you very much for pointing this out. We fully agree with this comment. Therefore, in this submission, we have made the necessary modifications to the expression of the statements. The changes were shown in line 579-581. Thank you again for your essential comment.

Round 2

Reviewer 2 Report

Comments and Suggestions for Authors

All requests were addressed promptly.